# Lycopene Inhibits Reactive Oxygen Species-Mediated NF-κB Signaling and Induces Apoptosis in Pancreatic Cancer Cells

**DOI:** 10.3390/nu11040762

**Published:** 2019-04-01

**Authors:** Yoonseon Jeong, Joo Weon Lim, Hyeyoung Kim

**Affiliations:** Department of Food and Nutrition, Brain Korea 21 PLUS Project, College of Human Ecology, Yonsei University, Seoul 03722, Korea; j_yoonseon@naver.com (Y.J.); jwlim11@yonsei.ac.kr (J.W.L.)

**Keywords:** apoptosis, NF-κB, lycopene, pancreatic cancer cells, reactive oxygen species

## Abstract

Generation of excess quantities of reactive oxygen species (ROS) caused by mitochondrial dysfunction facilitates rapid growth of pancreatic cancer cells. Elevated ROS levels in cancer cells cause an anti-apoptotic effect by activating survival signaling pathways, such as NF-κB and its target gene expression. Lycopene, a carotenoid found in tomatoes and a potent antioxidant, displays a protective effect against pancreatic cancer. The present study was designed to determine if lycopene induces apoptosis of pancreatic cancer PANC-1 cells by decreasing intracellular and mitochondrial ROS levels, and consequently suppressing NF-κB activation and expression of NF-κB target genes including cIAP1, cIAP2, and survivin. The results show that the lycopene decreased intracellular and mitochondrial ROS levels, mitochondrial function (determined by the mitochondrial membrane potential and oxygen consumption rate), NF-κB activity, and expression of NF-κB-dependent survival genes in PANC-1 cells. Lycopene reduced cell viability with increases in active caspase-3 and the Bax to Bcl-2 ratio in PANC-1 cells. These findings suggest that supplementation of lycopene could potentially reduce the incidence of pancreatic cancer.

## 1. Introduction

Pancreatic cancer is one of the leading causes of cancer deaths in the western world [1,2]. The likelihood of surviving this disease is extremely low worldwide as indicated by the 5-year survival rate of about 6% (ranges from 2% to 9%) [3]. In Korea, the 5-year survival rate of the pancreatic cancer was 8.8% in 2012 [4]. The ability of cells to evade apoptosis, a hallmark of human cancers, could be the reason for the strong resistance of pancreatic cancer to currently available treatments [5].

The currently available drugs in the market used for the treatment of pancreatic cancer are gemcitabine (2′,2′-difluorodeoxycytidine, GEM) and 5-fluorouracil (5-FU) [6,7]. These drugs exert their anticancer effects through the inhibition of DNA synthesis. In a single comparative study in patients with advanced pancreatic cancer, GEM was more effective than 5-FU with respect to survival duration and general clinical status [8]. Lee [9] recently reported that combination chemotherapy with 5-FU plus GEM yielded better survival rates than GEM monotherapy in pancreatic cancer patients. Like most anticancer agents, GEM and 5-FU induce ROS [10], which serve as additional anticancer mechanism. ROS induce various tissue damage [11]. Therefore, cells have to protect against cell damage by developing highly regulated antioxidant defense system. Ju et al. [12] reported that GEM-induced ROS stimulate the transcription of cytoprotective antioxidant genes, especially glutathione-generating enzymes. Increased antioxidant defense in GEM-treated cells make cancer cells resistant to GEM. Thus, the combination of an anticancer agent with antioxidant supplement has been suggested [13]. However, combination of 5-FU and antioxidant decreased the anticancer effect of 5-FU [14]. Therefore, prevention of pancreatic cancer by nutrient supplements or consumption of foods containing bioactive components becomes more important than cancer treatment by anticancer agents.

Although reactive oxygen species (ROS) are important regulators of normal cellular processes, uncontrolled generation of ROS contributes to the development of cancers. Owing to the accelerated metabolism, cancer cells have higher ROS levels than normal cells [15]. ROS facilitate cancer development by causing direct oxidative damage to DNA, and inducing lipid peroxidation or oxidizing proteins [16,17]. The major source of ROS production inside cells is the mitochondrial electron transport chain which generates electrons that can react with molecular oxygen [18].

The most well known metabolic abnormality in cancer cells is mitochondrial dysfunction, which renders the cells adapt glycolysis to generate adenosine triphosphate (ATP) even in the presence of oxygen, namely Warburg effect [19,20]. Warburg found that unlike normal cells, cancer cells tend to ferment glucose into lactate even in the presence of sufficient oxygen to support mitochondrial oxidative phosphorylation. Since energy requirements for cell proliferation are high, cancer cells complete catabolism of glucose using mitochondrial oxidative phosphorylation to maximize ATP production using Warburg effect [21]. Key enzymes implicated in the control of glucose metabolism and mitochondrial respiration are relatively highly expressed in cancers. Moreover, the expression levels of these key enzymes correlate with poor patient outcomes [22]. A large amount of evidence suggests that cancer cells exhibit increased intrinsic ROS stress, which enhances tumor metabolic adaptation, proliferation, survival and angiogenesis. Increased oxidative stress in cancer cells may be partly caused by mitochondrial dysfunction [23,24,25].

Elevated levels of ROS have anti-apoptotic effects in cancer cells that result from activation of redox-sensitive transcription factors such as nuclear factor κ-light-chain-enhancer of activated B cells (NF-κB). NF-κB activation leads to stimulation of cellular proliferation, cell migration and invasion, all of which contribute to carcinogenesis [26,27]. NF-κB, one of the key regulatory components in the anti-apoptotic signaling pathway, is a typical example of a transcription factor whose activity is subject to redox modulation [28,29]. In normal cells, NF-κB is localized in the cytosol in an inactive form bound to IκBα. Upon certain stimuli, IκBα is phosphorylated and dissociated from NF-κB for degradation. However, in cancer cells, NF-κB is maintained in an active form that promotes expression of pro-survival genes such as inhibitors of apoptosis (IAPs), which results in uncontrolled cell growth [30]. IAPs such as cIAP1, cIAP2, and survivin are involved in cell proliferation and survival, and are regulated at the gene level by activation of NF-κB [31]. Because NF-κB plays a pivotal role in mediating cell survival, blocking its NF-κB activity can alter the survival/death balance of tumor cells. Thus, targeting NF-κB might be an effective approach for developing treatments that combat human cancers.

To date, eight IAPs, which are intrinsic cellular inhibitors of apoptosis [32], have been identified. These include NAIP (BIRC1), c-IAP1 (BIRC2), c-IAP2 (BIRC3), X-linked IAP (XIAP, BIRC4), survivin (BIRC5), apollon (BRUCE, BIRC6), livin/ML-IAP (BIRC7) and IAP-like protein 2 (BIRC8). Among IAPs, cIAP1, cIAP2, and survivin promote the degradation of active caspase or block interactions between caspases and their substrates. A decrease in IAP levels induces apoptosis in pancreatic cancer cells [33,34]. Taken together, ROS induce NF-κB-mediated expression of IAPs in cancer cells. Therefore, reducing the generation and scavenging ROS by exogenous compounds in cancer cells should promote cancer cell death by decreasing IAPs.

Lycopene, an active component in tomatoes and other red vegetables, is a potent antioxidant and displays anticancer activities against various types of cancer such as prostate, pancreatic, breast and gastric cancer [35,36,37]. Epidemiologic studies have shown that a high intake of lycopene is associated with a reduced risk of pancreatic cancer [38,39,40,41]. In addition, genetic variations in the capacity to defend against oxidative stress and to repair oxidative DNA damage affect the risk of pancreatic cancer. However, some of these genetic predispositions can be alleviated by dietary antioxidants including lycopene [42]. Lycopene suppressed ROS-mediated cancer cell growth [43] and NF-κB nuclear binding in macrophages and SK-Hep-1 cells from a human hepatoma [44,45]. These studies suggest that lycopene has the potential beneficial effects in the prevention and treatment of pancreatic cancer through its inhibitory effect on cell proliferation promoted by ROS.

In the present study, we assessed the anti-cancer effect of lycopene on pancreatic cancer PANC-1 cells by determining its impact on cell viability, apoptotic indices (levels of active caspase-3 and ratio of Bax to Bcl-2), intracellular and mitochondrial ROS levels, and mitochondrial function (mitochondrial membrane potential (MMP) and oxygen consumption rate (OCR) as an index for mitochondrial oxidative phosphorylation). Activation of NF-κB, determined by measuring the level of IκBα phosphorylation and the NF-κB-DNA binding activity, and the expression of NF-κB target genes cIAP1, cIAP2 and survivin were also evaluated.

## 2. Materials and Methods

### 2.1. Cell Line and Culture Condition

Human pancreatic cancer cells (PANC-1) were obtained from American Type Culture Collection (Rockville, MD, USA) and maintained in Dulbecco’s Modified Eagle’s Medium (DMEM) medium (GIBCO, Grand Island, NY, USA) containing 4500 mg glucose/L and supplemented with 10% heat-inactivated fetal bovine serum (GIBCO) and antibiotic-antimycotic solution (100 U/mL penicillin and 100 μg/mL streptomycin). The cells were incubated at 37 °C in a humidified atmosphere of 5% CO_2_ and 95% air.

### 2.2. Experimental Protocol

PANC-1 cells (1–5 × 10^4^/mL) were treated with a solution of lycopene (L9879, Sigma-Aldrich, St. Louis, MO, USA) in tetrahydrofuran (THF) to produce final lycopene concentrations of 0.25 or 0.5 μM. The mixtures were then incubated for 24 h to determine cell viability, the levels Bcl-2 and Bax, intracellular and mitochondrial ROS levels, OCR, DNA binding activity of NF-kB and mRNA and protein expression of IκBα, cIAP1, cIAP2, and survivin. Controls for the experiments were PANC-1 cells incubated with THF (less than 0.3%) alone.

### 2.3. Determination of Cell Viability

The cells were seeded (1 × 10^4^/mL) in a 96-well plate and then cultured overnight. The cells were then treated with lycopene for 24 h. PANC-1 cells were incubated for 3 h with MTT (3-(4,5-dimethylthiazol-2-yl)-2,5-diphenyltetrazolium bromide; Sigma-Aldrich, St. Louis, MO, USA) in phosphate-buffered saline (PBS). The cells were lysed by mixing with 2-propanol in 0.1% HCl for 20 min using a shaker. The absorbances of the resulting mixtures were measured spectrophotometrically using a microplate reader (Molecular Devices, Sunnyvale, CA, USA).

### 2.4. Measurement of Intracellular Reactive Oxygen Species (ROS) Levels

The cells (5 × 10^4^/mL) were treated with lycopene for 24 h. For measurements of intracellular ROS, the cells were treated with 10 μg/mL of dichlorofluorescein diacetate (DCF-DA; Sigma-Aldrich) and incubated in 5% CO_2_/95% air at 37 °C for 30 min. The intensities of DCF fluorescence at 535 nm (excitation at 495 nm) were measured with a Victor 5 multi-label counter (PerkinElmer Life and Analytical Sciences, Boston, MA, USA). The intracellular ROS levels were normalized to cell numbers.

### 2.5. Measurement of Mitochondrial ROS Levels

The cells (5 × 10^4^/mL) were treated with lycopene for 24 h. To assess mitochondrial ROS levels, the cells were treated with 10 µM MitoSOX (Life technologies, Grand Island, NY, USA) for 30 min, before being washed and scraped into PBS. The intensity of MitoSOX fluorescence at 585 nm (excitation at 524 nm) was measured with a Victor 5 multi-label counter (PerkinElmer Life and Analytical Sciences). The mitochondrial ROS levels were normalized to cell numbers and thus, ROS levels were assessed based on equal cell numbers.

### 2.6. Measurement of Oxygen Consumption Rate (OCR)

OCR was assessed in real-time with a Seahorse XF96 Extracellular Flux Analyzer (Seahorse Bioscience, Billerica, MA, USA), which allows to measure OCR changes after sequential addition of modulators of respiration that target components of the electron transport chain in mitochondria. Cells (1 × 10^4^ cells/well/200 µL of DMEM) were plated in a XF 96 cell culture microplate (Seahorse Bioscience Inc., Billerica, MA, USA). Cells were washed with base media once, immersed in 175 µL base media, and incubated in the absence of CO_2_ for 20 min. After baseline measurements, respiration was measured after sequentially adding 25 µL of oligomycin (inhibitor of ATP synthase, 1 µg/mL), carbonyl cyanide-4 (trifluoromethoxy) phenylhydrazone (FCCP) (a protonophore and uncoupler of mitochondrial oxidative phosphorylation, 0.5 μM), and a combination of rotenone (mitochondrial complex I inhibitor, 1 μM) and antimycin A (mitochondrial complex III inhibitor, 1 μM) for OCR measurement using the XF Cell Mito Stress Test Kit (Cat. No. 103015-100, Seahorse Bioscience Inc., Billerica, MA, USA). OCR values were normalized for the protein content of each sample and expressed as the unit of pmoles/min. Basal OCR was expressed as percentage of the untreated cells (None). Basal OCR for “None” was set at 100.

### 2.7. Measurement of Mitochondrial Membrane Potential (MMP)

To determine changes in MMP, the cells were cultured on glass coverslips coated with Poly-L-lysine, pretreated with lycopene for 24 h. The cells were then incubated with 5,5′,6,6′-tetrachloro-1,1′,3,3′-tetraethyl benzimidazolyl carbocyanine iodide (JC-1) reagent (1:100; 10009908, Cayman Chemical Company, Ann Arbor, MI, USA) for 20 min. After removal of the media, the cells were dried for 15 min at room temperature, washed twice with PBS for 5 min, and mounted with mounting solution (M-7534, Sigma Aldrich, St. Louis, MO, USA). Fluorescence of JC-1 in the cells was determined (red; excitation at 590 nm and emission at 610 nm, green; excitation at 485 nm and emission at 535 nm) using a laser-scanning confocal microscope (LSM 880, Carl Zeiss Inc., Oberkochen, Germany). The fluorescence images were expressed as the percentage ratio of red and green intensities using NIH Image J 5.0 software (National Institutes of Health, Bethesda, MD, USA).

### 2.8. Real-Time Polymerase Chain Reaction (PCR) Analysis for cIAP1, cIAP2, and Survivin

The cells (5 × 10^4^/mL) were treated with lycopene for 24 h. Total RNA was isolated by using TRI reagent (Molecular Research Center, Inc., Cincinnati, OH, USA). Total RNA was converted into cDNA by using reverse transcription with a random hexamer and MuLV reverse transcriptase (Promega, Madison, WI, USA) and by heating at 23 °C for 10 min, 37 °C for 60 min and 95 °C for 5 min. The cDNA was used for real-time PCR with specific primers for cIAP1, cIAP2, survivin, and β-actin. The sequences of the cIAP1 primers used to produce the desired 160 bp PCR products are 5′-AGCTGTTGTCAACTTCAGATACCACT-3′ (forward primer) and 5′-TGTTTCACCAGGTCTCTATTAAAGCC-3′ (reverse primer). For cIAP2 cDNA production, the 160 bp PCR product was obtained by using the forward primer 5′-TCCTGGATAGTCTACTAACTGCC-3′ and reverse primer 5′-GCTTCTTGCAGAGAGTTTCTGAA-3′. Sequences of survivin primers were 5′-ATGGGTGCCCCGACGTT-3′ (forward primer) and 5′-TCAATCCATGGCAGCCAG- 3′ (reverse primer) to produce the desired 594 bp PCR product. For β-actin cDNA production, the 349 bp PCR product was obtained by using the forward primer 5′-ACCAACTGGGACGACATGGAG-3′ and reverse primer 5′-GTGAGGATCTTCATGAGGTAGTC-3′. For PCR amplification, the cDNA was amplified by 42 repeat denaturation cycles at 95 °C for 30 s, annealing at 58 °C for 30 s, and extension at 72 °C for 45 s. During the first cycle, the 95 °C step was extended to 3 min. The β-actin gene was amplified in the same reaction to serve as the reference gene.

### 2.9. Preparation of Cell Extracts

The cells (5 × 10^4^/mL) were treated with lycopene (0.25 and 0.5 µM) for 24 h. Cells were harvested by scraping with PBS, and pelleted by centrifugation at 5000× *g* for 15 min. The cell pellets were re-suspended with lysis buffer containing 10 mM Tris pH 7.4, 1% NP-40 and a commercial protease inhibitor complex (Complete; Roche, Mannheim, Germany), and the cells were lysed by drawing the suspension through a 1-mL syringe using several rapid strokes. The mixture was then incubated on ice for 30 min and centrifuged at 13,000× *g* for 15 min, giving supernatants that were collected and used as whole-cell extracts. To prepare the nuclear extracts, the cell pellets were re-suspended with 30 μL of hypotonic buffer, containing 10 mM 4-(2-hydroxyethyl)-1-piperazineethanesulfonic acid (HEPES) pH 7.9, 1.5 mM MgCl_2_, 10 m KCl, 0.5 mM DTT, 0.5 mM PMSF, 0.2% NP-40, and then placed on ice for 20 min. The extracts were centrifuged at 13,000× *g* for 20 min at 4 °C. The pellets were re-suspended in 30 μL of extraction buffer, containing 20 mM HEPES pH 7.9, 420 mM NaCl, 0.2 mM ethylenediaminetetraacetic acid (EDTA), 1.5 mM MgCl_2_, 25% glycerol, 0.5 mM DTT, 0.5 mM PMSF, and placed on ice for 20 min. The extracts were subsequently centrifuged at 13,000× *g* for 20 min at 4 °C, and the supernatants were used as the nuclear extracts. Protein concentrations were determined by using Bradford assay (Bio-Rad Laboratories, Hercules, CA, USA).

### 2.10. Western Blot Analysis for cIAP1, cIAP2, Survivin, IκBα, Bcl-2, and Bax

The cells (5 × 10^4^/mL) were treated with lycopene (0.25 and 0.5 µM) for 24 h. Aliquots from whole cell extracts were loaded onto 8%–12% sodium dodecyl sulfate (SDS) polyacrylamide gel (20–40 μg protein/lane) and subjected to electrophoresis under reducing conditions. The separated proteins were transferred to nitrocellulose membranes (Amersham, Inc., Arlington Heights, IL, USA) by electroblotting. Successful transfer of the proteins was verified using reversible staining with Ponceau S. The membranes were blocked using 3% non-fat dry milk in TBS-T (Tris-buffered saline and 0.2% Tween 20). The proteins were detected using antibodies for Bax (sc-526, Santa Cruz Biotechnology, CA, USA), Bcl-2 (sc-492, Santa Cruz Biotechnology, CA, USA), IκBα (sc-371, Santa Cruz Biotechnology, CA, USA), p-IκBα (#2859, Cell signaling Technology, Danvers, MA, USA), cIAP1(#AF8181, R&D systems), cIAP2(#MAB817, R&D systems), survivin (sc-10811, Santa Cruz Biotechnology, CA, USA) and actin (sc-1615, Santa Cruz Biotechnology) in TBS-T solution containing 3% dry milk, and incubated overnight at 4 °C. After washing with TBS-T, the primary antibodies were detected using horseradish peroxidase-conjugated secondary antibodies (anti-mouse, anti-rabbit, anti-goat), and using the enhanced chemiluminescence detection system (Santa Cruz Biotechnology). Actin served as a loading control. The ratio of Bax/Bcl-2 was determined from the protein-band densities of Bax and Bcl-2. The values are expressed as mean ± standard error (S.E.) of four different experiments.

### 2.11. Electrophoretic Mobility Shift Assay (EMSA)

The cells (5 × 10^4^/mL) were treated with lycopene for 24 h. Nuclear extracts (0.3 µg) of the cells were incubated with the ^32^P-labeled double-stranded oligonucleotide 5′-GGGCCAAGAATCTTAGCAGTTTCGGG-3 in buffer containing 12% glycerol, 12 mM Hepes (pH 7.9), 1 mM EDTA, 1 mM DTT, 25 mM KCl, 5 mM MgCl_2_, and 0.04 µg/mL poly[d(I-C)] at room temperature for 30 min. The extracts were then subjected to electrophoretic separation at room temperature on a non-denaturing 5% acrylamide gel at 30 mA using 0.5× Tris borate/EDTA buffer. The gels were dried at 80 °C for 1 h and exposed to radiography film for 24 h at −70 °C with intensifying screens.

### 2.12. Statistical Analysis

One-way analysis of variance (ANOVA), followed by Newman–Keul’s post hoc test, was used for statistical analysis. All data are reported as the mean ± S.E. of four different experiments. A *p*-value of 0.05 or less was considered to be statistically significant.

## 3. Results

### 3.1. Lycopene Induces Apoptosis in PANC-1 Cells

To determine the effect of lycopene on proliferation of PANC-1 cells, cell viability was measured using the MTT assay. As shown in Figure 1A, the viability of cells treated with lycopene decreased in a dose-dependent manner compared to that of untreated cells. To examine whether lycopene-induced decrease in cell viability is a consequence of an apoptotic effect, caspase-3 activation was determined by cleavage of pro-caspase-3 in lycopene-treated cells. Figure 1B showed that the level of the cleaved active form of caspase-3 increased in dose-dependent response to lycopene. Lycopene also promoted an increase in the level of pro-apoptotic Bax and a decrease in the level of anti-apoptotic Bcl-2 (Figure 1C), resulting in a dose-dependent increase in the Bax/Bcl-2 ratio.

### 3.2. Lycopene Decreases Intracellular and Mitochondrial ROS Levels and OCR in PANC-1 Cells

As shown in Figure 2A,B, lycopene caused decreases in intracellular and mitochondrial ROS levels in PANC-1 cells. To investigate whether mitochondrial function is affected in lycopene-induced apoptosis, OCR changes were measured. Real-time analysis of OCR changes was assessed by perturbing cells with modulators of respiration (oligomycin, FCCP, rotenone and antimycin A). Oligomycin is an inhibitor of ATP synthase (complex V). After injection of oligomycin, OCR decreased, which correlates to reduced mitochondrial respiration associated with cellular ATP production. FCCP is an uncoupling agent. After injection of FCCP, OCR is stimulated which can be used to calculate spare respiratory capacity. After a third injection of a mix of rotenone, a complex I inhibitor, and antimycin A, a complex Ill inhibitor, mitochondrial respiration decreased which enables the calculation of non-mitochondrial respiration. Therefore, monitoring the changes of OCR after injection of each modulator might be a good determination of mitochondrial function. All groups responded to these modulators in a similar way even though the OCR levels were different among groups (Figure 2C). Lycopene-treated cells showed relatively low OCR changes in response to the modulators compared to untreated cells. Figure 2D showed that basal OCR decreased in cells treated with lycopene, indicating that lycopene causes a reduction in mitochondrial function of PANC-1 cells. These results demonstrate that lycopene inhibits mitochondrial function and reduces mitochondrial and intracellular ROS levels in PANC-1 cells.

### 3.3. Lycopene Decreases MMP in PANC-1 Cells

JC-1 color change was used to determine MMP function. At lower energy state of mitochondria, JC-1 fluorescence is shown mainly as the green monomers, while higher MMP results in greater degrees of red JC-1 aggregates. We examined the ratio of aggregate and monomer fluorescence intensities in our confocal system by calibrating JC-1 fluorescence in PANC-1 cells. As shown in Figure 3, red and green fluorescence were found in untreated cells (“None”). Lycopene–treated cells showed prominent green fluorescence (“Lycopene”) as compared to “None”. JC-1 fluorescent color changed from red to green along with lycopene treatment, suggesting that lycopene induces MMP decline.

### 3.4. Lycopene Suppresses NF-κB Activation in PANC-1 Cells

In its resting state, NF-κB is bound with the inhibitor IκBα. Upon exposure to various stimuli, NF-κB activation occurs to release IκBα, which is then phosphorylated by IKK for proteasomal degradation. In addition, NF-κB is translocated to the nucleus where it transcribes target genes such as *cIAP1*, *cIAP2*, and *survivin*. In this study, we showed that DNA-binding activities of NF-κB are significantly reduced in lycopene-treated cells (Figure 4A). The inhibition of NF-κB by lycopene was confirmed using western blot analysis to assess the phosphorylation of IκBα. Lycopene inhibited phosphorylation of IκBα, indicating that it suppresses activation of NF-κB in PANC-1 cells (Figure 4B).

### 3.5. Lycopene Decreases Expression of cIAP1, cIAP2, and Survivin in PANC-1 Cells

To examine the effect of lycopene on expression of cIAP1, cIAP2 and survivin, PANC-1 cells were treated with lycopene after which the respective levels of mRNA and protein were determined. Cells treated with lycopene contained decreased levels of the transcribed mRNAs as well as the translated proteins (Figure 5A,B). These results demonstrate that lycopene induces apoptosis of PANC-1 cells by suppressing expression of cIAP1, cIAP2, and survivin, which are mediated by activation of NF-kB.

## 4. Discussion

In the present study, we found that lycopene induced apoptosis of pancreatic cancer PANC-1 cells by decreasing intracellular and mitochondrial ROS levels, and consequently inhibiting NF-κB activation and expression of NF-κB target genes such as cIAP1, cIAP2, and survivin. Lycopene also inhibited mitochondrial function, as indicated by decreases in MMP and basal OCR) in PANC-1 cells Because the NF-κB target genes (*cIAP1*, *cIAP2* and *survivin*) suppress caspase-3 activation, indirect inhibition of NF-κB target gene expression by lycopene promotes caspase-3 activation. Thus, lycopene induces caspase-3-dependent apoptosis and increased the Bax to Bcl-2 ratio in PANC-1 cells. The present finding is summarized in Figure 6.

ROS have been reported to be elevated in almost all cancers, and they appear to enhance cell survival and proliferation by stimulating carcinogenesis-related signaling pathways [46,47]. This is especially true in the development of pancreatic cancer where ROS serve as pro-survival and anti-apoptotic factors. An excess of ROS has been shown to have anti-apoptotic effects through activation of cancer-associated signaling pathways such as p38 mitogen activated protein kinases (MAPK), NF-κB and janus kinase/signal transducer and activator of transcription (JAK/STAT) [48,49,50,51,52,53]. These studies demonstrate that reducing ROS may induce the death of pancreatic cancer cells. It was supported by the present finding showing that decrease of ROS levels, by lycopene treatment, inhibited NF-κB-dependent expression of cell survival genes and thus, induced apoptosis in PANC-1 cells.

Regarding the effect of lycopene on mitochondrial function, Sandhir et al. [54] showed that 5-day administration of lycopene (10 mg/kg, orally) inhibited the activities of mitochondrial Complexes-II, IV and V along with reduction of mitochondrial respiration in striatum of 3-nitropropionic acid (3-NP)-treated rats. 3-NP caused neurobehavioral deficits with increased mitochondrial ROS and activated caspase-3 activity, which was suppressed by the administration of lycopene. Hantz et al. [55] showed that high concentrations of lycopene reduced mitochondrial membrane potential, an index of mitochondrial function, in prostate cancer cells. Therefore, mitochondria may be one of the target organs by lycopene treatment in some pathologic conditions including cancer. In the present study, lycopene decreased MMP, OCR, and mitochondrial ROS in PANC-1 cells, which supports the previous studies.

Fulda et al. [56] suggested that MMP decline can improve the efficiency of the cancer treatment. When the cells are disrupted by apoptosis–inducing stimuli, MMP changes occurs as the first event [57,58]. Therefore, maintaining MMP could prevent cancer cells from apoptosis [59]. In the present study, lycopene induced MMP decline, determined by color change from red to green. Untreated cells showed JC-1 aggregates (red) and monomers (green) while lycopene-treated cells have higher JC-1 monomers (green) than untreated cells. In PANC-1 cells, HT1080 human fibrosarcoma cells, and HepG2 hepatocellular carcinoma cells, JC-1 was distributed as aggregates (red) and monomers (green) without any treatment [60,61,62]. When MMP decreased by treatments, JC-1 monomers (green) increased and JC-1 aggregates (red) decreased. In the present study, Panc-1 cells showed both red and green fluorescence without lycopene treatment. Color was changed from red to green with lycopene treatment. Therefore, the fluorescence intensity ratio of red to green may be a more important indicator than red or green color intensity itself to determine MMP function.

In response to persistently high levels of ROS, the redox-sensitive transcription factor NF-κB is activated [63]. Constitutively activated NF-κB and elevated levels of ROS have been reported to make pancreatic cancer cells resistant to chemotherapy [64]. NF-κB exerts anti-apoptotic effects by inducing expression of IAPs [65]. IAPs are a group of anti-apoptotic factors that interact with each other by forming complexes which results in the suppression of protein or cooperative synergistic activities that protect cells from apoptosis [66,67,68]. IAPs are the only known endogenous proteins that suppress the activity of both initiator and effector caspases [69]. Among the IAP proteins, cIAP1, cIAP2 and survivin are known to be prominent members that exert anti-apoptotic effects by interfering with caspase-3, -7 and -9 [70,71]. Survivin reduces the auto-ubiquitination of IAPs, resulting in stabilization of IAPs and allowing interaction with caspase [72,73]. cIAP1 and cIAP2 are known to prevent downstream proteolytic processing of pro-caspase-3, -6 and -7 by blocking cytochrome c-induced activation of pro-caspase-9 in the intrinsic pathway [74]. Indeed, knockdown of cIAP1 and cIAP2 makes prostate cancer cells more sensitive to apoptosis [75] and knockdown of survivin inhibits cell growth in colorectal cancer and lung cancer cell lines [76]. These studies demonstrate that NF-kB-dependent genes play a critical role in inducing apoptosis by regulating caspases. The present study showed that lycopene suppressed NF-κB activation and expression of cIAP1, cIAP2 and survivin which was in parallel with a decrease in caspase-3 activation in PANC-1 cells. Lycopene also inhibits mitochondrial function, as indicated by decreases in OCR, and mitochondrial ROS in PANC-1 cells. Thus, lycopene-induced apoptosis appears to be associated with the reduction of intracellular and mitochondrial ROS and suppression of NF-κB signaling in pancreatic cancer cells.

Epidemiologic studies suggest that high consumption of fruits and vegetables, which are major sources of carotenoids, may play a role in the prevention of pancreatic cancer [77,78]. Drai et al. [79] showed that low concentrations of lycopene were shown in pancreatic cancer patients compared to normal health subjects. Abiaka et al. [80] demonstrated that low levels of lycopene and beta-carotene are strongly associated with a high risk of pancreatic cancer in Kuwait. A meta-analysis based publication from 1990 to 2013 showed that lycopene intake was inversely associated with pancreatic cancer risk, while alpha-carotene and cryptoxanthin intake had no significant relationship with pancreatic cancer risk [81]. They also reported that when stratified by ethnicity, there was an inverse relationship between lycopene intake and pancreatic cancer risk in Caucasians, while this correlation was not shown in the mixed population.

Kavanaugh et al. [82] described a U.S. Food and Drug Administration (FDA) review of tomato and/or lycopene intake with respect to risk reduction for certain cancers. The FDA found no evidence to support an association between lycopene intake and a reduced risk of pancreatic cancer. Nkondjock et al. [83] investigated the association between dietary carotenoids and pancreatic cancer risk using 462 pancreatic cancer cases and 4721 population-based controls in 8 Canadian provinces. They suggested that a diet rich in tomatoes and tomato-based products with high lycopene content may help reduce pancreatic cancer risk. Therefore, we have no concrete evidence between dietary intake of lycopene and reduction of pancreatic cancer incidence based on epidemiological studies. Since various factors, such as dietary factors, physical activity, smoking, genetic variations, ethnics, etc., are involved in epidemiologic studies, it may be difficult to show the direct relation of low lycopene intake and pancreatic cancer development.

The limitation of the present study is that we used one cell line (PANC-1 cells) for lycopene effect on expression of NF-κB-dependent survival genes and cancer cell survival. For a further study, more pancreatic cancer cell lines should be used to assess the anti-cancer effect of lycopene. More importantly, this should determine glucose consumption of the cells and medium content of lactate before and after lycopene treatment. This study may explain whether lycopene-induced decrease in OCR is a consequence of compensation by increase in anaerobic pathway or not.

Here, we found that lycopene inhibits oxidative stress and ROS-mediated NF-κB signaling pathway in pancreatic cancer cells. Since NF-κB target genes are survival genes (*cIAP1*, *cIAP2* and *survivin*) which suppress caspase-3 activation, lycopene induces caspase-3 - dependent apoptosis by suppressing survival genes in PANC-1 cells. Even though there is no concrete evidence of a link between dietary intake of lycopene and low risk of pancreatic cancer, we suggest that consumption of lycopene-containing foods or supplementation of lycopene may decrease risk of pancreatic cancer based on the present finding.

## Figures and Tables

**Figure 1 nutrients-11-00762-f001:**
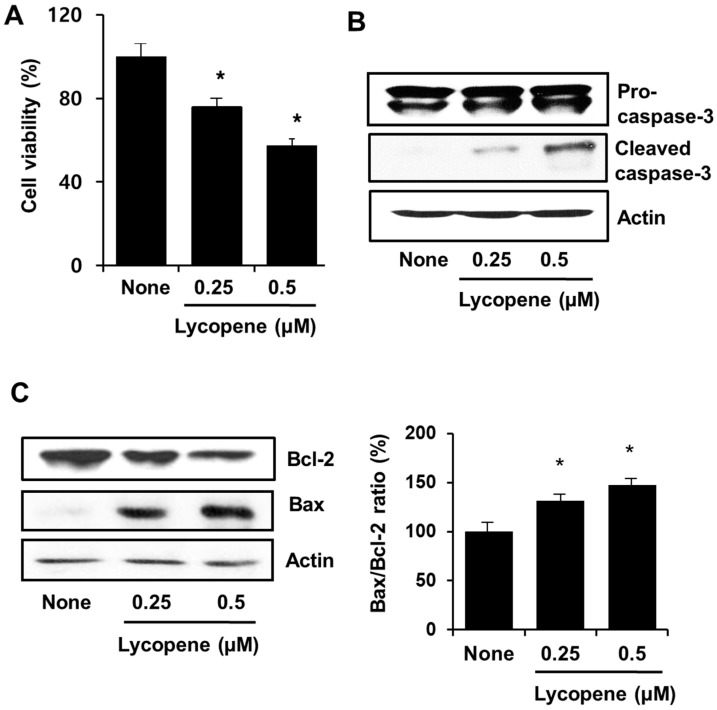
Lycopene decreased cell viability and increased caspase-3 activation and the Bax/Bcl-2 ratio in PANC-1 cells. Cells were treated with the indicated concentrations of lycopene for 24 h. (**A**) Cell viability was measured using the 3-(4,5-dimethylthiazol-2-yl)-2,5-diphenyltetrazolium bromide (MTT) assay. * *p* < 0.05 vs. “None”. None corresponds to the untreated cells; “0.25” and “0.5” correspond to the cells treated with 0.25 and 0.5 µM lycopene, respectively. (**B**) Caspase-3 activation was determined by the measuring the levels of pro-caspase-3 and cleaved caspase-3 in the cells. Designations of the columns are the same as in (**A**). (**C**) Levels of Bcl-2 and Bax in whole cell extracts were determined by using Western blot analysis. Designations of the columns are the same as in (**A**). The ratio of Bax/Bcl-2 was determined from the Bax and Bcl-2 protein band densities. * *p* < 0.05 vs. “None”. Designations of the columns are the same as in (**A**). The Bax/Bcl-2 ratio for “None” was set at 100.

**Figure 2 nutrients-11-00762-f002:**
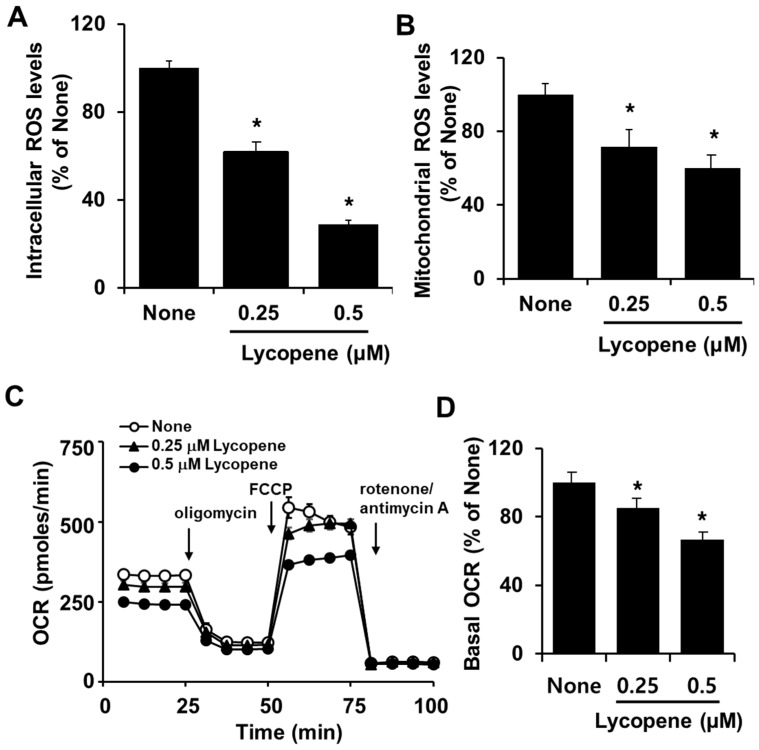
Lycopene decreased intracellular and mitochondrial reactive oxygen species (ROS) levels and oxygen consumption rate (OCR) in PANC-1 cells. Cells were treated with the indicated concentrations of lycopene for 24 h. (**A**,**B**) Intracellular and mitochondrial ROS levels were determined using dichlorofluorescein diacetate (DCF-DA) and MitoSoX, respectively. * *p* < 0.05 vs. the corresponding “None”. “None” corresponds to the untreated cells; “0.25” and “0.5” correspond to the cells treated with 0.25 and 0.5 µM lycopene, respectively. (**C**) The Seahorse XF96 analyzer was employed to assess OCR. Real-time analysis of OCR was assessed by perturbing cells with metabolic modulators (oligomycin, FCCP, rotenone and antimycin A). (**D**) Basal OCR was expressed as % of None. OCR for “None” was set at 100. * *p* < 0.05. Designations of the columns are the same as in (**A**).

**Figure 3 nutrients-11-00762-f003:**
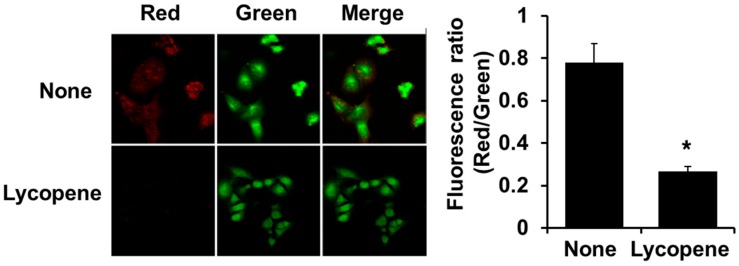
Lycopene decreased MMP in PANC-1 cells. Cells were treated with 0.5 µM for 24 h. The loss of MMP was examined using JC-1 staining. (**A**) Representative pictures are shown. JC-1 fluorescent dyes can gather in the matrix of mitochondria and produce red fluorescence. As the MMP decreased, JC-1 cannot gather the matrix and thus, JC-1 exists in the matrix as monomer, generating green fluorescence. (**B**) The ratio of the red and green fluorescence density is shown. Data are the mean ± S.E. of four different experiments. * *p* < 0.05 vs. the “None”. “None” corresponds to the untreated cells; “Lycopene” corresponds to the cells treated with 0.5 µM lycopene.

**Figure 4 nutrients-11-00762-f004:**
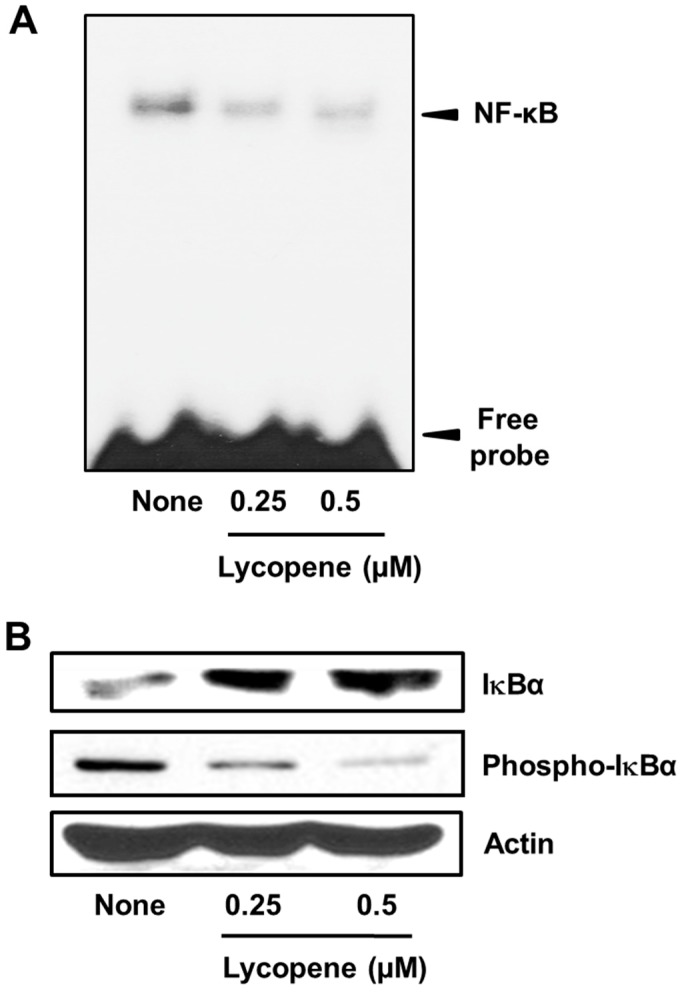
Lycopene reduced NF-κB-DNA binding activity and phospho-IκBα levels in PANC-1 cells. Cells were treated with the indicated concentrations of lycopene for 24 h. (**A**) DNA-binding activity of NF-κB in nuclear extracts was examined by using electrophoretic mobility shift assay (EMSA). “None” corresponds to the untreated cells; “0.25” and “0.5” correspond to the cells treated with 0.25 and 0.5 µM lycopene, respectively. (**B**) Western blot analysis was performed for the levels of phospho-IκBα and IκBα (and the protein standard actin). Designations of the columns are the same as in (**A**).

**Figure 5 nutrients-11-00762-f005:**
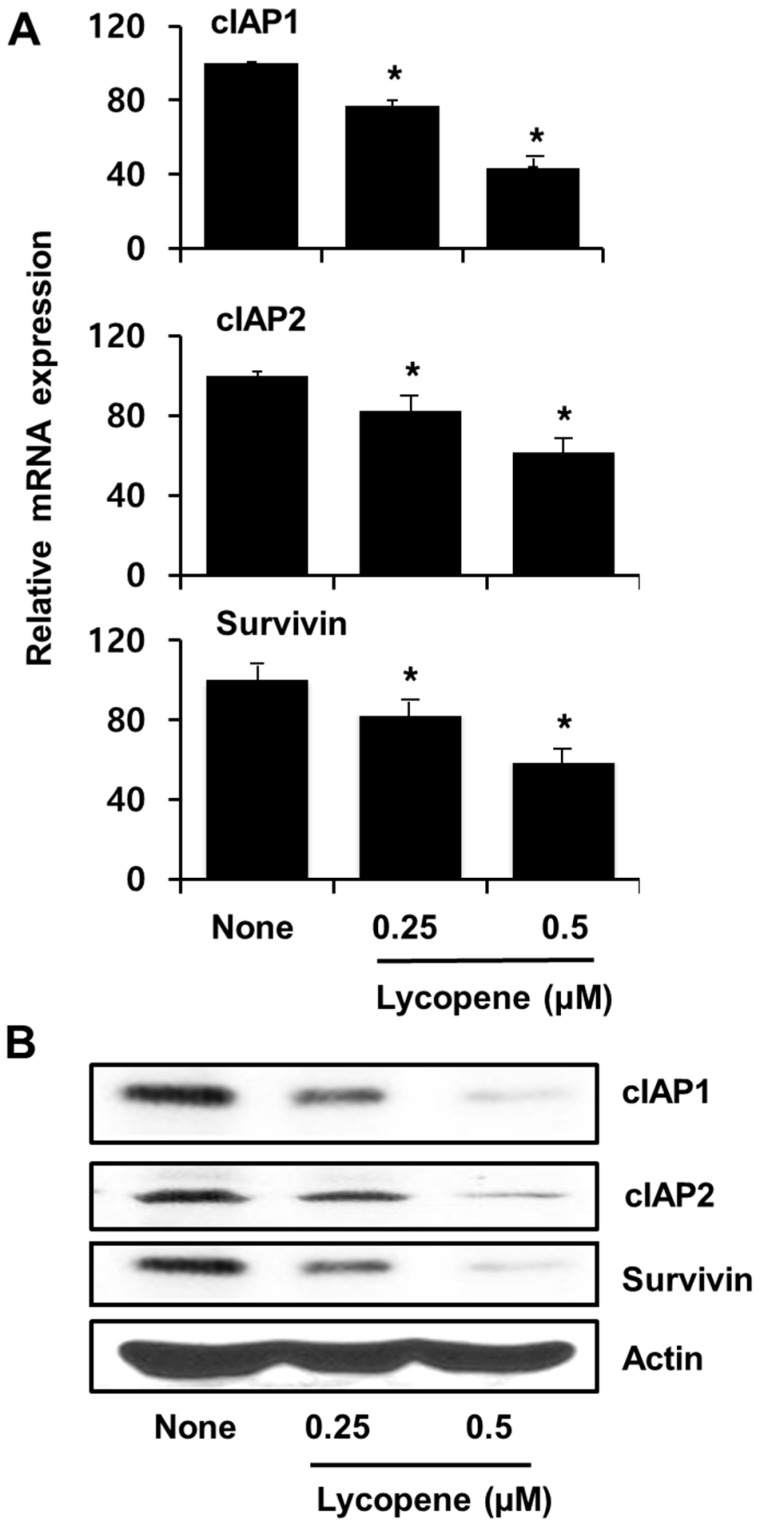
Lycopene decreased expression of cIAP1, cIAP2 and survivin in PANC-1 cells. Cells were treated with the indicated concentrations of lycopene for 24 h for mRNA and protein levels of cIAP1, cIAP2, and survivin. (**A**) mRNA levels of cIAP1, cIAP2, and survivin were determined by real-time polymerase chain reaction (PCR) analysis. * *p* < 0.05 vs. None. “None” corresponds to untreated cells, “0.25” and “0.5” to the cells treated with 0.25 and 0.5 µM lycopene, respectively. (**B**) Western blot analysis was performed for protein levels of cIAP1, cIAP2 and survivin (and the protein standard actin). Designations of the columns are the same as in (**A**).

**Figure 6 nutrients-11-00762-f006:**
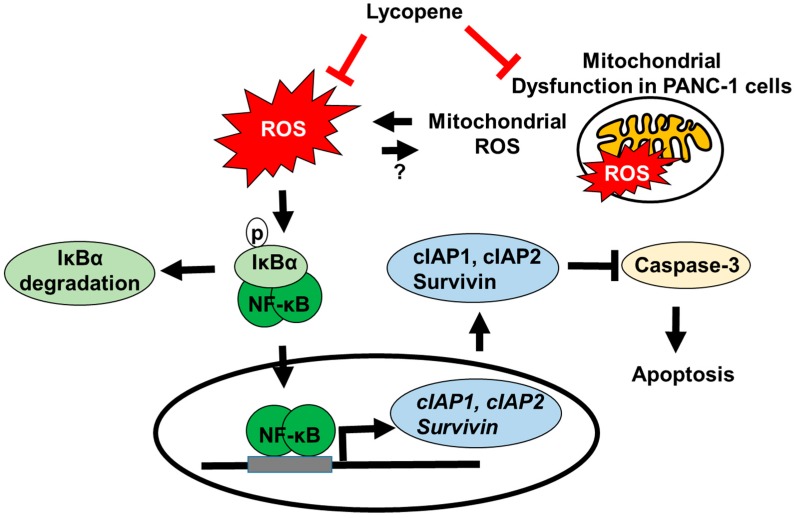
Schematic pathways for the inhibitory effect of lycopene on ROS-mediated NF-κB signaling and NF-κB target gene expression in PANC-1 cells.

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
