# Peer review of "Lycopene Inhibits Reactive Oxygen Species-Mediated NF-κB Signaling and Induces Apoptosis in Pancreatic Cancer Cells"

_nutrients, 2019, doi:10.3390/nu11040762_

Round 1

Reviewer 1 Report

The authors have sought to investigate the effect of lycopene on pancreatic cancer cells and determine the mechanism by which it promotes apoptosis of pancreatic cancer cells. They arrive at a conclusion that lycopene inhibits reactive oxygen species (ROS) which results in inhibition of anti-apoptotic effects of NF-KB signaling. The study overall is interesting and conducted well. There are, however, several comments that need to be address before the manuscript can be accepted. The english and language also needs to be improved significantly.

Comments:

1)     Introduction

a)     The authors need to point out what are the current available drugs in the market used for the treatment of pancreatic cancer and whether these have anti-oxidant properties.

b)     On line 37-38, do they mean uncontrolled??

c)     Paragraph 3- The authors need to talk about Warburg effect - emphasizing that most cancer cells produce energy primarily by a high rate of glycolysis followed by lactic acid fermentation in the cytosol, instead of by a comparatively low rate of glycolysis followed by oxidation of pyruvate in mitochondria as in normal cells

d)     In the concluding paragraph, you want to report your results briefly, not only talk about what was studied.

2)     Methods

a)     Were equal numbers of cells plated for the Seahorse experiment? This needs to be stated clearly.

b)     For the intracellular and Mitochondrial ROS experiments, equal numbers of cells must be seeded or the data shown should be normalized to cell number.

3)     Results

a)     For the intracellular and Mitochondrial ROS experiments, equal numbers of cells must be seeded or the data shown should be normalized to cell number. Because lycopene causes apoptosis, there are more cells in the control “none” wells compared to the 0.25/0.5um lycopene wells, hence fluorescence measured is proportionately low, hence this value must be normalized.

b)     The images for the mitochondrial MMP is not indicative of what is written in the text. The images does not show increase in green intensity in the lycopene treated cells. The lycopene treated cells only lack red fluorescence. Change this statement. Also why are healthy inapoptotic cells in the control “none” staining green? They should just stain red. Better images needed are also needed.

c)     For the OCR seahorse experiments, please elaborate what is plotted. Is it basal OCR? The authors also need to show the whole seahorse plot showing changes in OCR upon oligomycin, FCCP and rotenone/antimycin treatment, as they describe using all these reagents in their methods.

d)     Why did the authors choose 16h time point for the mRNA and protein analysis for survivin, cIAP1, cIAP2 whereas all the other experiments were conducted after 22h treatment. Authors should explain this.

4)     Discussion

a)     The discussion has too much background information on the effect on lycopene on the mitochondria and IAPs. These should be in the introduction section and not in the discussion. The discussion should relate to the results you observed to similar observations in other studies. The focus as written is on what was reported in prior studies and not connected to what you found

b)     The line 394 does not completely describe your findings- the results in the study point not only to mitochondrial ROS but also increased intracellular ROS that is resulting in NF_KB inhibition. Please elaborate

c)     The discussion should also point to deficiencies in your study and what are the future questions that need to be addressed.

5)     Figure legends

a)     Figure 1: legend title should be a brief summary of the results and not everything in the figure verbatim. Change -- “None”, where corresponds to “None”, which corresponds. Caspase – letter ‘e’ missing

b)     All figures legends micromolar sign “uM” missing next to 0.25 and 0.5

6)     English

a)     Line 39: Owing to their accelerated metabolism, cancer cells have increased ROS levels compared with those in normal cells. Remove “with those” and add to

b)     Line 50-51. “In addition, oxidative stress impairs mitochondrial function and propagates mitochondrial ROS production (17)”.  This sentence says the exact same thing as the previous sentence. Only keep one

c)     Line 82. “and some of these geneticly controlled effects can be modified by dietary antioxidants such as lycopene” – change to “Genetic predisposition can be alleviated by treating with lycopene”

d)     Line 88. “has potential beneficial effects of in the prevention and treatment of pancreatic cancer”. Remove “of”

e)     Line 242. “As can be seen by viewing the bar graph 242 shown in Fig. 1A” change to As shown in Fig. 1A

f)      Line 245-246. “by cleaving caspase-3 and pro-caspase-3” change to “by levels of cleaved caspase-3”

g)     Line 267. “Inspection of the graphs in Fig 2A and 2B shows that” change to “Fig 2A and 2B”

h)     Line 310. “The inhibition of NF-B by lycopene was confirmed by using western blot” Change “by using” to “using”

i)      Line 353. “ Supporting this proposal is the observation” change to “supporting this hypothesis”

j)  Line 373. “in the suppression of protein and cooperative synergistic”- Remove protein

Author Response

1)     Introduction

a)     The authors need to point out what are the current available drugs

Response: Author add it in line 36-51

b)     On line 37-38, do they mean uncontrolled??

Response: Author changed it to “uncontrolled” in line 53.

c)     Paragraph 3- The authors need to talk about Warburg effect

Response: Warburg effect in line 60-63.

d)     In the concluding paragraph:

Response: Authors reported results and concluded in line 442-448.

2)     Methods

a)     Were equal numbers of cells plated for the Seahorse experiment?

Response: Author add it in line 156.

b)     For the intracellular and Mitochondrial ROS experiments, equal numbers of cells must be seeded or the data shown should be normalized to cell number.

Response: It was described in line 138, 142, 145, 149-150.

3)     Results

a)     For the intracellular and Mitochondrial ROS experiments, equal numbers of cells must be seeded or the data shown should be normalized to cell number.

Response: It was described in line 138, 142, 145, 149-150. (as mentioned in methods)

b) healthy inapoptotic cells in the control “none” staining green?

Response: Authors described the reason why green color is shown in “None” in line 382-393.

c)     For the OCR seahorse experiments, please elaborate what is plotted. Is it basal OCR?

Response: Authors added plotted graph in Fig. 2C and described the results(line 279-287), Fig. 2 legend, and method (line 165).

Fig. 2D is basal OCR and it is described in 287-290 and Fig. 2 legend.

d)     Why did the authors choose 16h time point for the mRNA?

Response: Authors performed mRNA experiment at 16 h and 24h. Since the results are the same, authors revised the culture hour for mRNA exp., to 24h (line 124).

4)     Discussion

a)     The discussion has too much background information

Response: Authors rewrote the whole discussion based on the results of the present study.

b)     The line 394 does not completely describe your findings- the results in the study point not only to mitochondrial ROS but also increased intracellular ROS that is resulting in NF_KB inhibition. Please elaborate

Response: Authors rewrote the whole discussion and the relation of ROS and NF-kB is described in line 370.

c)     The discussion should also point to deficiencies in your study and what are the future questions that need to be addressed.

Response: Authors added it in line 436-441,

5)     Figure legends

a)     Figure 1: legend title should be a brief summary of the results and not everything in the figure verbatim. Change -- “None”, where corresponds to “None”, which corresponds. Caspase – letter ‘e’ missing

Response: Authors revised it.

b)     All figures legends micromolar sign “uM” missing next to 0.25 and 0.5

Response: Authors revised it.

6)     English

a)     Line 39: Owing to their accelerated metabolism, cancer cells have increased ROS levels compared with those in normal cells. Remove “with those” and add to

Response: Authors revised it (line 54).

b)     Line 50-51. “This sentence says the exact same thing as the previous sentence. Only keep one

Response: Authors removed this sentence.

c)     Line 82. “ “Genetic predisposition can be alleviated by treating with lycopene”

Response: Authors revised it. (line 97)

d)     Line 88. “has potential beneficial effects of in the prevention and treatment of pancreatic cancer”. Remove “of”

Response: Authors revised it. (line 100)

e)     Line 242. “As can be seen by viewing the bar graph 242 shown in Fig. 1A” change to As shown in Fig. 1A

Response: Authors revised it. (line 253)

f)      Line 245-246. “by cleaving caspase-3 and pro-caspase-3” change to “by levels of cleaved caspase-3”

Response: Authors revised it to “by cleavage of procaspase-3”. (line 256)

g)     Line 267. “Inspection of the graphs in Fig 2A and 2B shows that” change to “Fig 2A and 2B”

Response: Authors revised it. (line 274)

h)     Line 310. “The inhibition of NF-kB by lycopene was confirmed by using western blot” Change “by using” to “using”

Response: Authors revised it. (line 327)

i)      Line 353. “ Supporting this proposal is the observation” change to “supporting this hypothesis”

Response: Authors rewrote the whole discussion and removed this sentence.

j)  Line 373. “in the suppression of protein and cooperative synergistic”- Remove protein

Response: Authors rewrote the whole discussion and removed this sentence.

Reviewer 2 Report

The authors propose the possible inhibitory effect of lycopene on reactive oxygen species (ROS) in pancreatic cancer cells.

The experimental approach is simple, since they consider incubating pancreatic cancer cells in the presence of two doses of lycopene.

The authors seek effects of incubation on: cell viability, presence of ROS, oxygen consumption and expression of elements of the apoptotic pathway and of inhibitors of apoptosis.

Experiments show that the presence of lycopene produces:

Decrease in cell viability as a function of lycopene concentration

Decrease in the presence of ROS coupled with a decrease in oxygen consumption

Decreased mitochondrial permeability

Decreased expression of apoptosis inhibitors

Increased expression of Bax (pro-apoptosis) and the Bax / Bcl-2 ratio

All these data presuppose that the incubation of lycopene inhibits the functional capacity of the tumour cell, by acting at two levels: a first metabolic level, by decreasing its oxygen consumption, which leads to less formation of ROS and another level of gene expression, where the expression of the factors that inhibit apoptosis is inhibited.

Together, apoptosis is favoured and therefore the potential disappearance of these u cells.

Although the results seem to confirm the hypothesis that lycopene induces apoptosis in these tumour cells, it might have been better to offer a real measure of glucose consumption (which is supposed to be the main substrate), by the difference in concentration in the medium before and after incubation. It would also have been a definitive test, the measure of the capacity to form lactate, a fact that could rule out that the decrease in oxygen consumption is not compensated by the increase in the anaerobic pathway.

Minor points:

Methods:

The authors should specify what type of DMEM they have used: high or low glucose concentration.

Ln 112: Units are supposed to be μM

The authors should explain why they use different incubation times, 16 h for the expressions and 24 h for the rest of the determinations.

Results:

Caption of Figure 1: ln 257: units are μM?

                             Ln 258: Caspase-3

Caption of Figure 3: Ln 294: In apoptotic

Author Response

Although the results seem to confirm the hypothesis that lycopene induces apoptosis in these tumour cells, it might have been better to offer a real measure of glucose consumption (which is supposed to be the main substrate), by the difference in concentration in the medium before and after incubation. It would also have been a definitive test, the measure of the capacity to form lactate, a fact that could rule out that the decrease in oxygen consumption is not compensated by the increase in the anaerobic pathway.

 Response: Authors added it (line 438-441)

Minor points:

Methods:

The authors should specify what type of DMEM they have used: high or low glucose concentration.

Response: Authors revised it. (line 116)

Ln 112: Units are supposed to be μM

Response: Authors revised it. (line 123)

The authors should explain why they use different incubation times, 16 h for the expressions and 24 h for the rest of the determinations.

 Response: Authors performed mRNA experiment at 16 h and 24h. Since the results are the same, authors revised the culture hour for mRNA exp., to 24h (line 124).

Results:

Caption of Figure 1: ln 257: units are μM?

Response: Authors revised it. (line 265)

                             Ln 258: Caspase-3

Response: Authors revised it. (line 266)

Caption of Figure 3: Ln 294: In apoptotic

Response: Authors rewrote it. (Fig. 3 caption)

Round 2

Reviewer 1 Report

The authors have addressed all my suggestions. I recommend acceptance of the manuscript